# Pharmacogenetic Dose Modeling Based on CYP2C19 Allelic Phenotypes

**DOI:** 10.3390/pharmaceutics14122833

**Published:** 2022-12-16

**Authors:** Julia Carolin Stingl, Jason Radermacher, Justyna Wozniak, Roberto Viviani

**Affiliations:** 1Institute of Clinical Pharmacology, University Hospital of RWTH, 52074 Aachen, Germany; 2Institute of Psychology, University of Innsbruck, 6020 Innsbruck, Austria; 3Psychiatry and Psychotherapy Clinic, University of Ulm, 89075 Ulm, Germany

**Keywords:** pharmacogenetics, dose adjustment, cytochrome P450 enzymes, drug metabolism, CYP2C19, horseshoe, random effects, shrinkage

## Abstract

Pharmacogenetic variability in drug metabolism leads to patient vulnerability to side effects and to therapeutic failure. Our purpose was to introduce a systematic statistical methodology to estimate quantitative dose adjustments based on pharmacokinetic differences in pharmacogenetic subgroups, addressing the concerns of sparse data, incomplete information on phenotypic groups, and heterogeneity of study design. Data on psychotropic drugs metabolized by the cytochrome P450 enzyme CYP2C19 were used as a case study. CYP2C19 activity scores were estimated, while statistically assessing the influence of methodological differences between studies, and used to estimate dose adjustments in genotypic groups. Modeling effects of activity scores in each substance as a population led to prudential predictions of adjustments when few data were available (‘shrinkage’). The best results were obtained with the regularized horseshoe, an innovative Bayesian approach to estimate coefficients viewed as a sample from two populations. This approach was compared to modeling the population of substance as normally distributed, to a more traditional “fixed effects” approach, and to dose adjustments based on weighted means, as in current practice. Modeling strategies were able to assess the influence of study parameters and deliver adjustment levels when necessary, extrapolated to all phenotype groups, as well as their level of uncertainty. In addition, the horseshoe reacted sensitively to small study sizes, and provided conservative estimates of required adjustments.

## 1. Introduction

Pharmacogenetic variability in drug metabolism represents an important type of patient vulnerability brought about by differences in individual drug clearance, with the consequence of an increased risk of side effects and therapeutic failure [1]. The influence of pharmacogenetic polymorphism in drug metabolism can be compensated for by adapting the dose to receive equalized concentration-time curves in individuals with different pharmacogenetic subtypes. Pharmacogenetic patient-stratification and -dosing has just recently been applied in clinical drug development [2]. Pharmacogenetic clinical guidelines summarizing the pharmacogenetic literature and evidence are increasingly used in clinical practice, but for most drugs on the market these guidelines (such as those issued by the Clinical Pharmacology Implementation Consortium, CPIC), give qualitative recommendations rather than proposing quantitative dose adjustments [3,4]. However, in clinical practice, when dealing with a patient with a certain pharmacogenetic profile, quantitative dose-adjustments may be the most practical tool to avoid vulnerability and personalize treatment while allowing the administration of the drug or continued therapy.

Developed to address this need, pharmacogenetic dose recommendations provide quantitative estimates of dose adjustments based on mean differences in oral clearance between genotype groups extracted from evidence from clinical studies [5,6,7]. However, pharmacogenetic dose recommendations have been suffering from methodological shortcomings deriving from the small sample-sizes of individual studies. The uncertainty and possible overestimation of effects induced by small sample-sizes have been exacerbated by computing the dose adjustment in each phenotype group and substance separately. The purpose of the present work is to evaluate a statistical approach to estimate dose adjustments from multiple studies on different substrates, to quantify and counter uncertainty introduced by small sample-sizes and study heterogeneity.

The adoption of a statistical model may have several advantages. First, a statistical model allows the efficient pooling of information from studies. In current pharmacogenetic dose adjustments, such as those developed by Kirchheiner et al. (2004), these adjustments are based on weighted means computed on each sampling point (a combination of phenotype and substance) [6]. In that work, the weights take effect across studies within each sampling point, but have no influence between sampling points. Instead of estimating the mean effects of pharmacogenetics in these phenotypes–substance combinations, we will estimate the extent to which a substrate is affected by a pharmacogenetic polymorphism across all phenotypes (poor, intermediate, extensive, rapid and ultrarapid metabolizer-phenotype), and derive the adjustment score from the combination of this estimate and the metabolic activity of the phenotype (“activity score”). For example, if one allele (activity score: −1) leads to a 20% decrease in standard dose, a homozygous genotype with two equal alleles (activity score: −2) should lead to a 40% decrease, assuming linearity of the effects of activity scores on adjustments. The verification of the linear effect of activity scores on dose adjustments needs to be preliminarily carried out from the data.

Second, by modeling all data together, it becomes possible to estimate the effect of different characteristics of the original studies and adjust for possible effects of their methodological diversity. Pharmacokinetic studies differ in several respects, such as the dosage used, the parameter on which the pharmacokinetic adjustment is computed, and the sample type (patients or healthy volunteers). It is of interest to be able to estimate the effect of these differences in the execution of the study, hopefully showing them to be of limited magnitude.

Third, a statistical model may provide an assessment of the uncertainty of the estimated amount of dose adjustments. This provides one guarantee against issuing dose adjustment recommendations without sufficient empirical data in their support. A further guarantee may be given by using statistical estimation techniques that set coefficient estimates to prudential values when these coefficient estimates are large and were obtained from small samples. Note that these techniques still allow for making efficient use of sampling points from small samples and including them in pooled estimates, while avoiding putting too much faith in their individual coefficient estimates. In contrast, when large amounts of information exist for a specific substance, the estimate will be made with more confidence.

We will also assume this dataset to be composed of many studies where the drug is a substrate of CYP2C19, and a smaller number of studies of an exploratory character, where there is no evidence for metabolism by CYP2C19. This requires allowing a strong degree of heterogeneity in the effect sizes of CYP2C19 polymorphism on drug metabolism in the dataset. Unlike a meta-analysis, which assumes that one and the same effect was investigated with variations in methods and sample recruitment, we may not assume here that all studies investigated the same effect.

To achieve these multiple aims, we explored the use of a Bayesian approach (the regularized horseshoe, [8]) to provide conservative estimates of dose adjustments and assess their merit based on sufficient evidence for individual drugs. This dataset will be used here more as a test case to develop methodology than as a dataset in which sufficient information exists to provide reliable guidance for clinical decision making.

The regularized horseshoe was chosen here because of the multiple requirements for our model. First, the regularized horseshoe may provide prudential estimates of adjustments, through its sophisticated implementation of the shrinkage of coefficient estimates. ’Shrinkage’ is a technical statistical term that refers to biasing individual coefficient-estimates towards a conservative value when the evidence for these estimates is low. This may be particularly advantageous for drugs where few data exist. Second, it may allow for greater heterogeneity than other approaches.

To evaluate the performance of shrinkage and the heterogeneity of the regularized horseshoe, we will compare it with the outcome of other more traditional modeling strategies. In the first, we will obtain shrinkage by modeling the coefficients as normally distributed. This modeling strategy is related to standard meta-analytic approaches, which model heterogeneity between studies as a normally distributed random-effect. In comparison, the horseshoe may be viewed as replacing this normal distribution with a Cauchy (which may be viewed as a distribution with unknown variance). In the second, the effects of phenotype and individual drugs on the pharmacokinetic parameters will be modelled as the ‘fixed effect’ of the interaction between phenotype and substance. In this modeling strategy, data collected for one substance have little influence on estimates of another, and the relative estimates are essentially separate. Therefore, this modeling strategy allows for the largest degree of heterogeneity between estimates, but may suffer from giving too much faith to large estimates from small samples.

## 2. Materials and Methods

### 2.1. Literature Search

Study search and inclusion: we performed a search on all the psychotropic drugs that have been included in prior meta-analyses and reviews [3,4,5,6,7,8] on CYP2C19 pharmacogenetic effects on pharmacokinetic parameters. The research terms “drug” and “CYP2C19” were used to select the relevant studies from the literature in PubMed. Subsequently, the studies were selected that provided data in humans on dose-related pharmacokinetic parameters (such as clearance, area under the concentration-time curve, AUC, drug concentrations at steady-state normalized for dose, Css) separately for the CYP2C19 metabolizer phenotypes. In the older literature, mephenytoin phenotyping was used to discriminate between poor metabolizer and normal metabolizer. Since the poor metabolizer phenotype (PM) as characterized with mephenytoin corresponds to the homozygous *CYP2C19*2* or **3* allelic phenotypes, these were both considered as PM. Intermediate metabolizers (IM) were defined as carriers of one high-activity allele (*CYP2C19*1*) and one inactive allele (*CYP2C19*2 or *3*). The rapid metabolizer phenotype (RM) was defined as carrier of one *CYP2C19*17* allele in combination with *CYP2C19*1*. The ultrarapid phenotype (UM) was defined by the *CYP2C19*17/*17* genotype. Single-case reports were not included into the database.

### 2.2. Data Selection

Dose adjustments may be estimated from percent differences in oral clearance, as reported in the pharmacokinetic studies for different metabolizer groups defined by *CYP* genotype. By assuming that a given population is characterized by certain distributions of pharmacogenetic metabolizer types, estimation of percent dose-adjustment can be computed from a standard dosage of 100 percent in the overall population, bearing in mind that this latter consists of a mixture of poor metabolizers, intermediate, rapid and ultrarapid metabolizers, with known mixing rates [6]. Differences in pharmacokinetic parameters proportional to dosage may be calculated as percent differences in oral clearance. These may be subsequently transformed into dose adjustments in as far as the clearance is directly proportional to the effective dose (in conditions of linear kinetics).

The approach used so far to calculate dose adjustments consists of computing these estimates separately for each study and metabolizer group, and estimating for each metabolizer group an average percent dose-adjustment as the mean of the adjustments computed from the study data.

Here, we entered these dose adjustments into a database (see Table 1 and Appendix A) for subsequent modeling (see Section 2.2.1, below). In addition, we noted into the database the properties of studies that may have conceivably affected these estimates, such as pharmacokinetic parameters (AUC, CL, Css), single or multiple dose, healthy participants or patients, and the pooling of genotypes that would fall into another CYP2C19 phenotype group (see parameters listed in Table 1). 

#### 2.2.1. Statistical Analysis

We considered in this study four models for the heterogeneity of the coefficients: “fixed” effects, modeling the coefficients as a Gaussian distribution, and two versions of the continuous-shrinkage horseshoe: non-regularized and regularized. The basic model takes the form
(1)y=Xβ+Rθ+ε
where **y** is here the vector of the dose adjustments reported in the literature (each sampling point being the adjustment of a phenotype–substance combination given by a study), and **X** is a set of *j* = 1, 2, … *D* predictors, on whose coefficients **β** the prior exerts variable selection and shrinkage. These predictors were here the CYP2C19-activity scores (centered on the EM phenotype) for each substance, modeling a linear effect of activity scores on adjustments, one per substance. These coefficients are the focus of the analysis, since they determine the recommended dose adjustments. **R** is a set of confounding predictors, whose coefficients, **θ**, are estimated outside the Gaussian or the horseshoe prior, i.e., without shrinkage, here set to properties of studies whose effects we wanted to adjust for. The error term, **ε**, modelled residual variability between reported adjustments in the phenotype–substance combinations of the individual studies. Here, we refer to these combinations in each study as “sampling points”, since they represent individual observations in the model.

Several properties of this model deserve comment. First, the model had no constant term. The adjustments computed from the literature are computed by comparing pharmacokinetic data between an allelic group and the EM phenotype. The adjustment for the EM phenotype in this computation is zero, as it would be given by the comparison of the data of the EM phenotype with itself. Depending on the context, we refer below to this baseline value as “zero adjustment” or “100% adjusted dose”. Omitting the constant term constrains the data to zero adjustment in the EM group. Second, data for the EM phenotype are not independent data, as they are used in the literature to compute the adjustments for the other phenotypes. Hence, these data were not used in the model. Third, no predictor coded the main effect of activity scores, so that the coefficients of the activity scores × substances coded in the matrix of predictors **X** represented the effect of activity scores in each substance, instead of the deviations from a main-activity-score effect. As a result, these coefficients were allowed to shrink to zero. In contrast, a model with the main effect of activity scores shrinks these coefficients toward this main effect. When including a main effect, when there are few data in support of the role of CYP2C19 in the metabolism of a given substance, the model predicts an effect approximately equal to this main effect. This was not desirable, since, as noted, we wished to avoid the assumption that all studies in the literature reported data on substances that are metabolized by CYP2C19. Instead, because the extent of shrinkage is determined from the data, we wanted the models to carry out inference about which substances carried CYP2C19 effects larger than zero by seeking evidence relative to no effects. Finally, the model contained no random effect for studies because study variation may be captured by the estimate of the variance between the individual sampling points given within the studies. Furthermore, the effect of activity scores in several substances was estimated by one study only, so that variation at the higher level is modelled by the Gaussian or horseshoe prior itself.

In the “fixed effects” model, the coefficients **β** and **θ** were given uninformative priors. All other models kept the same uninformative priors for the confounder coefficients **θ,** but differed in the prior for **β**. In the standard random-effect model, the prior for **β** was given by
(2)$$\beta_{j}|\tau ∼N\left(0,\tau^{2}\right),\;\tau ∼C^{+}\left(0,1\right)$$
i.e., a Gaussian distribution with variance *τ*^2^, which was given in turn a weakly informative half-Cauchy hyperprior. Alternative distributions, such as a Student’s *t* with 3 or 4 degrees of freedom, were explored in the present work, but since they gave results that differed very little from the Gaussian prior, they were not pursued further.

The prior for **β** in the horseshoe [57] and its regularized version used here [8] is the means through which these models implement Bayesian variable selection and shrink coefficient estimates toward zero:$$\beta_{j}|\tau ,\lambda_{j}∼N\left(0,\tau^{2}\xi_{j}^{2}\right)\;\tau ∼C^{+}\left(0,0.1\right),\;\xi_{j}∼C^{+}\left(0,1\right)$$

The variance of the Gaussian prior for the coefficients, **β**, is given be the product of two terms. The first term *τ* is a global shrinkage parameter. With small estimates of this parameter, all coefficients are shrunk towards zero, because of its multiplicative effects in the above expression. The second term, *ξ_j_*, takes a different value for each coefficient *β_j_* and allows the set of these coefficients to be heterogeneous. These parameters are given half-Cauchy hyperpriors through *λ_j_* to allow for outliers modeling substances with little or no CYP2C19 metabolism.

In the typical application of the horseshoe, the combined effect of the two terms in the variance of the Gaussian is that of a sparsity prior. The horseshoe approximates to a model in which the coefficients are drawn from two normally distributed populations, one of which realizes values close to zero [58]. Hence, these typical applications of the horseshoe show its capacity to shrink small coefficients in models with many predictors, to near-zero values. Here, in contrast, we used the horseshoe to model heterogeneity based on the results of [8], which showed that the prior for *τ* may be used to encode a wide range of prior rates of non-zero parameters. We modelled *τ* with a positive half-Cauchy prior, implying a 50% positive coefficient rate. This scale parameter is the most sensitive parameter of the prior model, as it is related to an estimate of positive coefficient rates; in Appendix A, we conducted a sensitivity analysis to verify its influence.

The regularized horseshoe replaces *ξ_j_* above with
$$\xi_{j}^{2}=\frac{c^{2}\lambda_{j}^{2}}{c^{2}+\tau^{2}\lambda_{j}^{2}}\;\lambda_{j}∼C^{+}\left(0,1\right),\;c∼\mathrm{Student}\text{-}t\left(0,2.5,df=8\right)$$

The parameter *c* further regularizes the coefficients, thus preventing large coefficient values arising from small samples. For small *c*, this parameter moderates the size of the individual coefficients. When $c^{2}\gg \tau^{2}\lambda_{j}^{2}$, the regularized horseshoe reverts to the non-regularized form, where $\xi_{j}∼C^{+}\left(0,1\right)$ [8]. Full Bayesian inference for *c* is provided through a Student’s *t* distribution (here with df = 8) and a hyperprior scale parameter.

In most studies, no information was available on the variability of estimates. However, sample sizes were provided. These varied widely (from *N* = 1 to *N* = 507), so that it was essential to keep this information into account. For this reason, a Bayesian meta-analytic approach was adopted to weight sampling points according to the reported sample size. The sampling point variability was modelled as
(3)$$\varepsilon_{i}|\sigma_{w},\sigma_{b}∼N(0,\frac{1}{n_{i}}\sigma_{w}^{2}+\sigma_{b}^{2}),\;\sigma_{w}∼\log\text{-}\mathrm{normal}\left(4,0.25\right),\;\sigma_{b}∼\log\text{-}\mathrm{normal}\left(4,0.25\right)$$
where $n_{i}$ was the known sample size of each sampling point *i*, whereas $\sigma_{w}^{2}$ and $\sigma_{b}^{2}$ were the parameter variance within, and the variance between, the sampling points to be estimated from the data, respectively. The variance between sampling points provides a bound on the precision of parameter estimates of studies of increasing sample size; its effect was to increase the credibility intervals of estimates of effects of activity scores for which large studies were available. To identify these parameters in this relatively small sample, we provided informative log-normal priors based on the plausible variability of this kind of measurements. These priors ensured that both variance parameters were sampled away from zero, thus avoiding the instability given by one parameter picking up all variance and leaving the other at zero. It is worth noting that, although these informative priors suppressed this instability, estimates of other parameters and their intervals in the model were not much affected by it, since the weighting of sampling points was determined by the weighted sum of both terms (when one went up, the other went down, giving similar sums). For this reason, the informativeness of this prior does not result in artificially narrow credibility intervals of dose adjustments. Information on the parameters used to fit these models and samples from the posterior for important coefficients are in Appendix A.

Before fitting the data to this model, we conducted preliminary analyses on the data, to establish values of activity scores for the genotype, to verify the linearity of the effects of activity scores, and to identify the properties of studies affecting the estimates. These preliminary analyses used models with a set of predictors with uninformative priors as “fixed” effects, a random effect of studies, and sampling-point variability modelled as the same sum of variance components as in the main model. Further information on these models is in Appendix A.

All models were fitted with Stan, a package implementing Hamiltonian Monte Carlo [59] through the R interface. The Stan code of these models is given in Appendix B.

## 3. Results

### 3.1. Selection of Studies

The total number of studies retrieved for information on *CYP2C19* genotype-dependent clearance/AUC or Css data was N = 52. One study was excluded because it did not conform to the declaration of Helsinki on good clinical practice. The study details are given in Table 1. Sixteen studies used phenotyping as the method for CYP2C19 phenotype assessment, and all the remaining studies used genotyping. The **17* allele was determined in N = 18 studies; phenotypes predicted from all genotypes were available in N = 11 studies. 

Earlier studies that used phenotyping substrates only discriminated between PM and EM. The PM determined by phenotyping corresponds to a genotype consisting of homozygous alleles with zero CYP2C19 enzyme activity (either coded by the **2* allele or the **3* allele, denoted as the **null/*null* genotype). The IM group consists of one active and one inactive allele (**1/*null*) of *CYP2C19*. In the case of the **17* allele, the **17/*null* genotype was at times grouped together with the **17/*17* genotype in the IM phenotype. This is reported in Table 1 as ‘pooling’, since the majority of the (earlier) studies (N = 28) performed before the identification of *CYP2C19*17* in the year 2006 [60] did not discriminate **17/*null* from **1/*null* genotypes. One study pooled the PM genotype (homozygous **2* carriers) into the IM group [33]. While the RM group is defined as genotype **1/*17*, five studies did not discriminate between the UM and RM groups, and pooled possible carriers of the homozygous **17* genotype into one group. 

Variations in study design comprised the pooling of certain genotype groups into one phenotype group (e.g., the pooling of **17/*1* and **17/*17* alleles into the EM, UM or RM group), pharmacokinetic outcomes (such as AUC, clearance or Css), studies carried out on healthy volunteers versus patient studies, and studies on single-dose kinetics or multiple-dose kinetics. In the following modeling section, these variations will be considered separately, based on whether they may affect the estimate of dose adjustments in individual phenotypic groups or the estimate of adjustments generally, across groups. The former will be considered in the section on estimates of activity scores that follows, and the latter in the rest of the modeling.

### 3.2. Estimate of CYP2C19 Activity Score from Dose Adjustments

We preliminarily estimated models of average activity-score, to verify the linearity of its effect on dose adjustments reported in the literature. For the estimation of the CYP2C19 activity predicted by allelic combinations of the **null* (**2* or **3*), the **17*, and the wildtype allele, dose adjustments were plotted as a function of phenotype, coded in activity-score values spaced at equal intervals from the PM to the UM phenotypes (Figure 1A).

Several properties of the data are apparent from this plot. First, the fitted line, which is a cubic polynomial, shows the effects of the phenotypic groups to be almost linear. There was a slight inflection in the middle of the curve, suggesting the effects of the RM and UM groups to be slightly smaller than those of the IM and PM groups. Second, the individual points are spread upwards, relative to the fitted line, suggesting that the modeling might profit from a transformation of the input data. Finally, some studies with small sample sizes provided very strong outliers. One study on mianserine (by Dahl et al., 1994, [46]) reported an adjustment that strongly contradicts physiological expectations, increasing the dose by approximately 45% in PMs, instead of decreasing it.

For use in the models of the next section, we estimated activity scores from the data in three steps. First, we modelled the mean dose adjustments as the coefficients of a factorial model, with the phenotype groups as levels. To weigh the sampling points for sample size, we estimated residual variability with the model expressed in Equation (3). We excluded studies where no CYP2C19 effect was visible, i.e., drugs that are not metabolized by CYP2C19 in vivo (mianserine, maprotiline, fluoxetine, and fluvoxamine).

Here, we looked at whether or not phenotypic groups that were investigated while pooling diverse genotypes, affected estimates of dose adjustments. As one may expect, studies pooling homozygous **17* carriers into the RM group overestimated the effects; the large value of this overestimation, 38% (95% credibility interval, 11–66%), was the difference in dose adjustment between these studies and those that did not pool (the studies that pool were small, and averaged an adjusted dose of 156%, while those that did not pool gave an adjustment of 115%). It should be noted, however, that the credibility intervals of this effect were large, reflecting the uncertainty of its effective size, due to the small number of studies that were affected by it. In contrast, there were more studies that included **17* carriers in the IM group. However, there was no detectable effect of this inclusion (2%, credibility interval −16–18%).

We therefore opted to estimate activity scores after excluding studies that pooled the **17* homozygous carriers into the RM group, to avoid the possible bias arising from this pooling. We set the baseline adjustment to the EM group (which was then given a score of zero), and the scale to the estimated adjustment of the IM group (which was given a score of −1), as this group has one **2* allele (note that the base and scale of a measure are chosen by convention; we based the scale on the IM group because there were more data in this group than in the PM and RM groups). The results of this analysis are displayed in Table 2.

By averaging the estimates of the allelic effects in the RM and the UM groups, we obtained an estimate of 0.8 for the activity score of one **17* allele. This gave activity scores where PM (genotype **null/*null*) had an approximate score of −2, RM (genotype **1/*17*) of 0.8, and UM (genotype **17/*17*) of 1.6. The fit of the adjustment data on the phenotypic groups, spaced according to the new activity scores, is displayed in Figure 1B. These results are broadly consistent with current assumptions on CYP2C19 activity scores [15].

In a further step, we verified the absence of deviations from linearity by testing a different slope for the **17* carriers in the dataset as a whole, and by adding quadratic and cubic polynomials to the activity-score predictor. None of these model additions were significant (further information and details on these models are in Appendix A).

Finally, we evaluated the need of a logarithmic transformation of the data. When considering a log-transformation of the adjustments, we obtained an effect that was no longer linear for equally spaced activity-scores (Appendix A). In addition, an examination of residuals with these activity scores showed no asymmetry (Appendix A). We concluded that these activity scores could be used in the models that follow, and that no logarithmic transformation was required on the input data.

### 3.3. Effects of Study Properties on Estimated Dose Adjustments

We considered here study properties that may affect the estimate of the slope of the activity-scores effects as a whole. In some studies, the EM allelic group was defined while pooling it with other alleles, the **2* or the **17*. This pooling affects all other phenotypic groups, because the EM estimate is the reference point in the original studies, affecting all other phenotypes at once. Pooling heterozygous **2* in the EM group (which happened in the old phenotyping studies that pooled all genotypes other than **null/*null* into one group of EM), had no effect on the estimate of activity scores in our data. Instead, the pooling of **17* carriers in the EM group (**1/*17* and **17/*17*), which happened in the older studies that genotyped for the **null* alleles but not for **17* alleles, led to smaller effects of activity scores on adjustments in the UM phenotype (−16% per activity-score point). However, these effects failed to reach significance (see Appendix A).

We also tested the possible effect of the pharmacokinetic measurement used to estimate the dose adjustments. Some studies (usually performed on healthy participants as a pharmacokinetic study), used AUC or clearance, while other studies (usually on patients and in naturalistic conditions), used Css, the dose-corrected plasma concentration at steady state. We therefore tested the influence of the pharmacokinetic parameter given in the studies, and the of participants type (healthy versus patients). Studies using AUC or clearance revealed a slightly lower (but not significant) slope, compared with the studies with Css (4% to 8% differences in adjustments per activity-score point, see Appendix A). In these studies, saturation at higher doses may lead to nonlinear kinetics and biased estimates of the CYP2C19 metabolic pathway. However, another possible source of bias is that these studies, which were more recent, were conducted prevalently on substances where the role of CYP2C19 was already known, and therefore do not include substances where the effect is small. 

The study property single- versus multiple-dose did not result in significant differences in the estimation of the CYP2C19 activity effect on dose. Instead, we investigated the effect of this study property on the within-study variance, which was lower in studies with single dose. However, the large credibility intervals of these estimate did not justify the inclusion of this study property in the model (see Appendix A).

For modeling the substance-specific dose adjustments, only the factor pooling of **17/*17* carriers to the RM group was considered, because this had a significant effect on the estimation of the dose adjustments in this group, as shown in the previous section, and this effect was estimated to be large.

### 3.4. Modelled Dose Adjustments

We used the activity scores derived in the previous section to model adjustment as a function of phenotype (Equation (1)), adding the pooling of homozygous **17/*17* carriers into the RM phenotype group as a confounding covariate. In this model, the coefficient of the activity score gives the adjustment for these phenotypes relative to the label dose, i.e., the slope of this adjustment. These slopes estimate differences in CYP2C19 affinity for the substrate, the importance of the metabolic pathway, and other unmeasured sources contributing to the variation in adjustment levels distributed across a population of substances, of which the substances in the study are a sample.

We first compared the four approaches, to estimate these coefficients slopes, described in the Methods section: “fixed”, a Gaussian random-effect, and the two versions of the horseshoe (non-regularized and regularized). In Figure 2, panel A, the estimated adjustments obtained with these four methods are displayed, together with their credibility intervals. One can see that the fixed-effects approach gives the most extreme adjustments, while the horseshoe approaches provide the most conservative estimates, with litte differences between both approaches. However, the horseshoe without regularization was more difficult to fit (the Stan engine gave warnings that estimates may not be reliable). The Gaussian random-effect provides estimates located in-between the fixed and horseshoe approaches.

When there was evidence from many studies, and with large sample sizes, the adjustments (as is the case for escitalopram) were estimated with small confidence intervals and were nearly identical in all the four methods. The conservativeness of the Gaussian random-effect and of the horseshoe affected estimates when they were based on few or single studies, and sample sizes were small. For example, etizolam, doxepine and diazepam were given adjustment estimates that were almost as large as those of escitalopram by the fixed-effect approach, but all shrinkage approaches reacted sensitively to small numbers of studies and small sample sizes, and moved estimates and credibility intervals towards zero.

Panels B and C of Figure 2 show the action of the two mechanisms through which statistical modeling deviated from the previous approach of dose adjustments, calculated as weighted means of the adjustments of the single studies (substance-specific phenotype-group means). The estimates from the weighted-mean approach are shown in orange, while the adjustments computed with the fixed approach (panel B) and the regularized horseshoe (panel C) are given by the points of the fitted slopes that correspond to the phenotype groups on the x axis. Both models (fixed-effect or random-effect) deviate from the earlier approach, because estimating one slope of activity scores instead of computing separate estimates in each phenotype group, pools information from all groups. The consequence is most apparent in the case of amitriptyline, where the UM-adjustment estimates of the fixed effects and the weighted-mean approach differ. However, the fixed-effects approach gives large adjustment estimates for small sample sizes, as shown for diazepam and etizolam. In constrast, these estimates are shrunken towards zero by the horseshoe, where they are also affected by sample size. Detailed information and diagnostics for these models are in the Appendix A.

## 4. Discussion

In this work, we estimated pharmacogenetic dose adjustments for psychotropic drugs based on clinical pharmacokinetic data, to account for differences in drug exposure caused by the *CYP2C19* polymorphism. In contrast to earlier approaches, where mean values provided an overview on the quantitative decrease or increase in drug clearance detected in the pharmacogenetic CYP2C19 phenotype groups (the means model), we used a statistical modeling approach to better account for the uncertainty caused by a paucity of data (due to small sample sizes and few studies, leading to missing values for genotype groups), and possible methodological confounds across studies in estimated dose adjustments. Our aim was to provide dose adjustments based on a model where the lack of data, the heterogeneity of the studies, and the uncertainty of phenotype–genotype correlation is included and used for a conservative estimation. 

We compiled all the existing data for this study, entering the former dose adjustments [7], together with those from more recent studies for all psychotropic drugs where the *CYP2C19* genotype was characterized. We used this dataset to estimate an activity score for the CYP2C19 phenotype predicted from the genotype. This estimation approach resulted in roughly three groups of drugs: those where a strong effect of the CYP2C19 polymorphism was evident (escitalopram, citalopram, sertraline and venlafaxine), those where a tendency was detected but the uncertainty estimation would not allow the issuing of a quantitative dose adjustment (the tricyclics, clozapine, etizolam, diazepam, moclobemide), and those where the data did not support any effect of CYP2C19 (mianserine, maprotiline, fluoxetine, fluvoxamine, zotepine; see Figure 2A). The drugs with the strongest influence are also mentioned by other reviews on the pharmacokinetics of CYP2C19 [61], but no quantitative dose adjustments have been issued so far. 

### 4.1. Consequences of Basing Estimates of Pharmacogenetic Effects on Estimates of Pathways: Comparison with Existing Models

The formulation of guidelines for pharmacogenetic dose adjustments has been the purpose of previous works of ours and of others [3,4,5,6,7,61]. However, this existing work has estimated dose adjustments based on the means model, i.e., the evidence from the comparison of single genotype groups in comparison with the EM group. This model suffers from several shortcomings. One of theses is exemplified by cases where it was judged that there was enough evidence to issue dose recommendations for one phenotypic group, but not for another. However, the extent to which a pharmacogenetic polymorphism may affect the metabolism of a compound, depends on it being part of the metabolic drug pathway. Hence, evidence from the pharmacogenetic effects in one phenotypic group implies the existence of effects in the others. The efficiency of the model can be improved by including evidence for all phenotype groups at once into one model estimating the slope of the increase in drug clearance dependent on the CYP2C19 activity score. Because an activity score is a predictor of genetic effects on enzyme activity, the dose adjustments may be predicted, based on the slope of the activity estimated for a given drug, extrapolating from this slope for any pharmacogenetic subgroup, and even for metabolic activity levels that lie in-between those of the genetic groups, such as those determined by phenotyping [62]. For example, the estimation of CYP2C19 dose adjustments for clozapine is based on data for PM, IM, and RM, while there are no data on homozygous **17/*17* (UM). Dose recommendations based on the means model will conclude that there is no evidence for making a recommendation for the UM phenotype, due to lack of data for this group. However, the evidence on the pathway is present from the other groups. Since the slope of that activity score can predict the amount of dose adjustments for all phenotype groups of CYP2C19, dose adjustments for all phenotypes may be computed for all drugs where data on clearance differences in at least two phenotypes are available.

In addition to insufficient study data, there are several limitations challenging the validity of dose recommendations based on pharmacogenetics. One major issue is the variance in published studies. In our final model, we assumed uniform within-study variance, which was estimated based on the sample size of each sampling point. However, differences in pharmacogenetic designs may also result in systematic differences in the precision of estimates [63]. For example, an observational study on TDM data may have a large sample size (such as the several hundreds of individuals per genotype group of the studies by Jukic et al.) [36], whereas a pharmacogenetic panel study performed on genotype-selected healthy volunteers matched by age, body mass index and dose, and measured at equal study conditions, may lead to precise estimates, even with very small sample sizes.

### 4.2. Consequences of Modeling Differences between Study Methodologies

A further concern arises from systematic methodological differences in earlier, smaller studies. Since early studies were based on phenotyping before the discovery of genetic variants, the historical evolvement of pharmacogenetic knowledge has gradually changed the classification of phenotype groups. We tried to address the concern of methodological heterogeneity by formally testing the effects of pooling allelic groups into one phenotype. However, the outcomes of earlier studies may have influenced which substances were investigated later on, resulting in a publication bias. For example, if an early small study failed to detect an influence of the CYP2C19 polymorphism on a substance, this drug may not have been followed up eventually by more recent studies applying more sophisticated pharmacogenetic methods. In our analysis, we verified the fact that methodological differences were not so large as to affect the outcome of the analysis directly. However, the bias due to substances being no longer actively investigated shows up in large credibility intervals.

In our analysis of the CYP2C19 activity score, we estimated a smaller slope of the *CYP2C19*17* allele compared with the opposite effect of the **null* allele. This is in concordance with previous reports on the review of the evidence, suggesting that the magnitude of effect of the *CYP2C19*17* allele is considerably smaller than that of *CYP2C19*2* and *CYP2C19*3* [15,64]. The **17* allele is a variant allele, leading to higher enzyme expression and activity which, however, may be affected by numerous concomitant individual factors. The higher expression and activity may not be the same in all individuals and in all tissues, including the liver. When direct comparisons are made, the magnitude of the effect of the *CYP2C19*17* allele is considerably smaller than that of *CYP2C19*2* and *CYP2C19*3* [64]. Our model may therefore overestimate the activity scores of the **17* allele, due to publication bias. 

### 4.3. Statistical Methodology to Address Variability in the Amount of Data Available for Substances

An important feature of the present work is the adoption of shrinkage approaches, which were compared to a “fixed effects” approach and to the traditional weighted-mean approach. Relative to the traditional approach, all others make more efficient use of the available information if activity scores are available, since they use information from all phenotypic groups simultaneously. Relative to fixed effects, shrinkage introduces prudential estimates when these latter are made on the basis of few data. The advantages offered by shrinkage are widely recognized in the statistical community (see for example [65] and discussion), but its practical application has been confined to a few areas. For pharmacogenetic dose adjustments, we found that shrinkage of any form reacted senstitively to estimates based on few data, and may be preferable to fixed effects estimates. We consider the regularized horseshoe to be the preferred approach, as it appears to provide the most prudential estimates. Furthermore, the Gaussian random-effect approach may be somewhat misspecified, as the coefficients, being heterogeneous, may not be normally distributed. The horseshoe may accommodate heterogeneity better, as it was developed to fit a mixture of two distributions of random coefficients.

An argument for the adoption of linear-adjustment estimates and shrinkage of a comprehensive statistical model is that the combination of estimates and credibility intervals may give an actionable summary of the information on dose adjustments. In panel A of Figure 2, we can see that we have enough data to provide quantitative dose adjustments on a handful of substances, based on their narrow credibility intervals. Perhaps even more importantly, for several substances there was clear evidence of a CYP2C19 metabolic pathway, but the estimation of the effect of the polymorphism was based on so little data that at present a quantitative dose adjustment cannot be formulated, calling for further research.

## 5. Conclusions

Pharmacogenetic dose recommendations are a practical tool to address patient vulnerability and improve the benefit–risk ratio of therapy. We used a statistical modeling approach to better account for the uncertainty caused by the paucity of data and methodological confounds across studies. We compared different approaches to address this uncertainty, and found out that, relative to fixed effects, shrinkage delivered estimates that were more prudential when the latter were made on the basis of little data. Modeling approaches provide a rational basis for formulating quantitative dose adjustments in personalized treatment. This may facilitate the administration of drugs or the continuation of therapy in patients with vulnerabilities due to pharmacogenetic risk-profiles. 

## Figures and Tables

**Figure 1 pharmaceutics-14-02833-f001:**
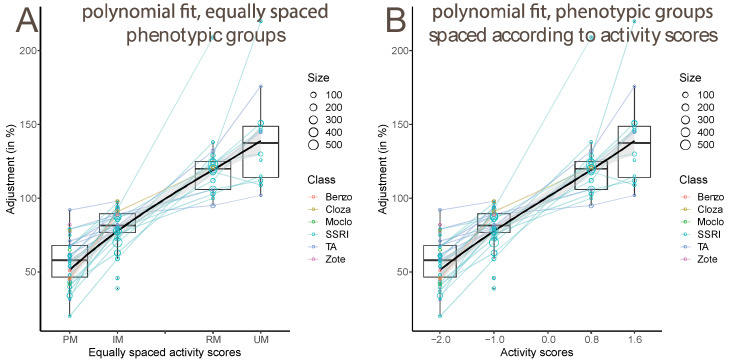
(**A**) Estimated dose adjustments were plotted as a function of phenotype, coded in expression grades of equal intervals from the PM to the UM phenotypes. The fitted line is a cubic polynomial, and shows the effect to be almost linear, with a slight inflection downwards in the middle. (**B**) The same plot, with the phenotypic groups spaced at the intervals of the estimated activity scores. The fitted line is a cubic polynomial again, this time showing a linear fit.

**Figure 2 pharmaceutics-14-02833-f002:**
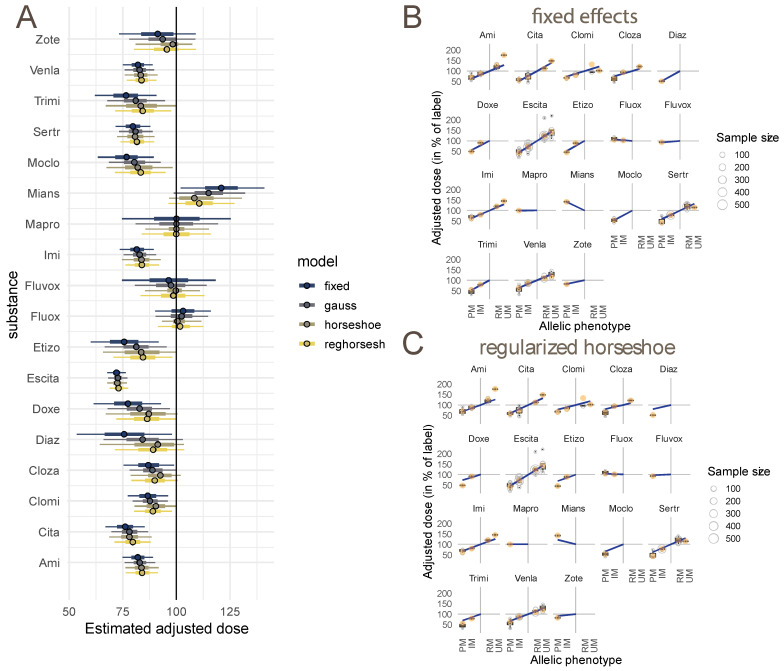
(**A**) Comparison of estimated dose adjustments and confidence intervals for the IM phenotype in the fixed-effects model, the Gaussian random-effect, and the two versions of the horseshoe. (**B**) Fixed-effects model: slopes of effects of activity scores in individual substances (blue), together with estimates of adjustment computed with the traditional approach of means computed at each combination of phenotype group and substance, separately (in orange). On the x axis, the four phenotypic groups for which adjustment was estimated; on the y axis, estimated adjusted dose (in % of label dose). (**C**) As in B, regularized horseshoe model.

**Table 1 pharmaceutics-14-02833-t001:** Studies included in the analysis and their characteristics.

Drug	Parameter	Study Subjects/Dosing	Dose	Identification of Metabolizer Groups	Reference
Amitriptyline	Css	Patients, MD	25–225 mg	*CYP2C19*2, CYP2C19*3*	[9]
Amitriptyline	Css	Patients, MD	150 mg	*CYP2C19*2*	[10]
Amitriptyline	AUC	Healthy, SD	50 mg	*CYP2C19*2*	[11]
Amitriptyline	Css	Patients, MD	150 mg	Phenotyping	[12]
Amitriptyline	Css	Patients, MD	100–150 mg	*CYP2C19*2, CYP2C19*17*	[13]
Amitriptyline	AUC, MR	Patients, MD	25 mg	*CYP2C19*2, CYP2C19*3*	[14]
Amitriptyline	AUC	Healthy, SD	25 mg	*CYP2C19*2, CYP2C19*17*	[15]
Clomipramine	Css	Patients, MD	10–250 mg	*CYP2C19*2, CYP2C19*3*	[16]
Clomipramine	1/CL	Healthy, SD	100 mg	Phenotyping	[17]
Clomipramine	Css	Patients, MD	25–300 mg	*CYP2C19*2, CYP2C19*17*	[13]
Doxepine	1/CL	Healthy, SD	75 mg	*CYP2C19*2*	[18]
Imipramine	1/CL	Healthy, SD	100 mg	Phenotyping	[19]
Imipramine	Css	Patients, MD	70 mg	Phenotyping	[20]
Imipramine	Css	Patients, MD	50 mg	Phenotyping	[21]
Imipramine	Css	Patients, MD	dose adjusted	*CYP2C19*2*	[22]
Trimipramine	Css	Patients, MD	350 mg	Phenotyping	[23]
Trimipramine	1/CL	Healthy, SD	75 mg	*CYP2C19*2*	[24]
Citalopram	AUC	Healthy, MD	40 mg	Phenotyping	[25]
Citalopram	Css	Patients, MD	10–60 mg	*CYP2C19*2, CYP2C19*17*	[13]
Citalopram	CL	Healthy, SD	20 mg	*CYP2C19*2*	[26]
Citalopram	Css	Patients, MD	35 ± 20/34 ± 17	*CYP2C19*2*	[27]
Escitalopram	Css	Patients, MD	4.8–7.4 mg	*CYP2C19*2, CYP2C19*17*	[28]
Escitalopram	CL	Healthy, MD	10 mg	Phenotyping	[29]
Escitalopram	AUC	Healthy, SD	5 mg	*CYP2C19*2, CYP2C19*17*	[30]
Escitalopram	Css	Patients, MD	16 ± 5/21 ± 13	*CYP2C19*2*	[31]
Escitalopram	Css	Patients, MD	20 ± 9/22 ± 10	*CYP2C19*2*	[27]
Escitalopram	Css, MR	Patients, MD	5–40 mg	*CYP2C19*2, CYP2C19*17*	[32]
Escitalopram	Css	Patients, MD	10–20	*CYP2C19*2, CYP2C19*17*	[33]
Escitalopram	Css	Patients, MD	10 mg	*CYP2C19*2, CYP2C19*17*	[34]
Escitalopram	Css	Patients, MD	5–20 mg	*CYP2C19*2, *3, CYP2C19*17*	[35]
Escitalopram	Css	Patients, MD	12.6–18.1 mg	*CYP2C19*2, CYP2C19*17*	[36]
Escitalopram	MR	Patients, MD	10–80 mg	*CYP2C19*2, CYP2C19*17*	[37]
Fluoxetine	AUC, CL	Healthy, SD	40 mg	*CYP2C19*2, CYP2C19*17*	[38]
Fluoxetine	Css	Patients, MD	10–60 mg	*CYP2C19*2, CYP2C19*17*	[39]
Fluvoxamine	AUC	Healthy, SD	100 mg	Phenotyping	[40]
Sertraline	AUC, CL	Healthy, SD	100 mg	Phenotyping	[41]
Sertraline	CL	Patients, MD	dose–adjusted	*CYP2C19*2, CYP2C19*17*	[42]
Sertraline	Css	Patients, MD	dose–adjusted	*CYP2C19*2, CYP2C19*17*	[43]
Sertraline	AUC	Healthy, SD		*CYP2C19*2, CYP2C19*17*	[44]
Maprotiline	Css	Patients, MD	150 mg	Phenotyping	[45]
Mianserin	AUC	Healthy, SD	30 mg	Phenotyping	[46]
Moclobemide	1/Cl	Healthy, MD	300 mg; 600 mg	Phenotyping	[47]
Venlafaxine	Css	Patients, MD	<225 mg; >225 mg	*CYP2C19*2, CYP2C19*17*	[48]
Venlafaxine	Css	Patients, MD	dose–corrected TDM	*CYP2C19*2, CYP2C19*17*	[49]
Venlafaxine	Css	Patients, MD	150 mg	*CYP2C19*2, CYP2C19*17*	[50]
Clozapine	AUC	Healthy, SD	10 mg	Phenotyping	[51]
Clozapine	Css	Patients, MD	250 (25–800) mg	*CYP2C19*2, CYP2C19*17*	[52]
Clozapine	Css	Patients, MD	433 mg	*CYP2C19*2, CYP2C19*17*	[53]
Zotepine	1/CL	Healthy, SD	25 mg	Phenotyping	[54]
Etizolam	AUC	Healthy, SD	1 mg	*CYP2C19*2, *3*	[55]
Diazepam	CL	Healthy, SD	10 mg	Phenotyping	[56]

**Table 2 pharmaceutics-14-02833-t002:** Estimates and 90% credibility intervals of activity scores after excluding RM pooling.

Estimate	EM	IM	PM	RM	UM	*CYP2C19*17* ^+^
Median	0	−1	−1.96	0.70	1.75	0.79
5% lower	0	−1	2.91	0.26	1.13	0.48
95% upper	0	−1	−1.42	1.28	2.76	1.26

+ The last column is the estimate of the activity score of one **17* allele obtained by averaging the estimates of the effect of one allele computed in the RM and UM groups.

## Data Availability

Data used in this study are available in the Appendix A. The procedure is implemented in R code to be called from RStudio. The Bayesian model specifications are called by R directly. The RStudio markdown and Stan files of all models are made available as Appendix A, and could be immediately used with new datasets.

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
