# Peer review of "Pharmacogenetic Dose Modeling Based on CYP2C19 Allelic Phenotypes"

_pharmaceutics, 2022, doi:10.3390/pharmaceutics14122833_

Round 1

Reviewer 1 Report

Very interesting paper for both pharmacological and methodological point of view. I have only a note: Stan is a very specialistic software, diffused in the pharmacogenetic community. If the authors want to reach a wider community, I suggest, for the future, to implement the procedure in a language that is more diffused, such as Python or R.

Author Response

Reviewer 1 wrote: Very interesting paper for both pharmacological and methodological point of view. I have only a note: Stan is a very specialistic software, diffused in the pharmacogenetic community. If the authors want to reach a wider community, I suggest, for the future, to implement the procedure in a language that is more diffused, such as Python or R.

Answer: We would like to thank the reviewer for her appreciative words and her suggestion.

The procedure is implemented in R code to be called from RStudio. The Bayesian model specifications are called by R directly. The markdown and stan files of all models are made available as supplementary material and could be immediately used with new datasets. We now included this explanation in the data availability statement.

General answer:

We would like to thank the reviewers for their reading of our manuscript and their comments. 

In addition to these changes we

  • corrected a mistake in the numbering of the sections (two sections had the same numbering 3.1, instead of 3.1 and 3.2)
  • made minor changes to the discussion to improve readability, including the introduction of subtitles to facilitate location of discussion of specific issues.

Reviewer 2 Report

This paper elaborates on earlier approaches on PGx based dose adjustments in the area of neuropsychpharmacology. I do have some concerns that should be implemeneted prior publication: 

1. the beauty of advanced statistical modelling is that you always get a result. However, without any validation one never knows how accurate the results are. It is correct that a few methods have been conducted as sensitivity analyses, however, this per se does not provide proof of accuracy that should the authors provide. 

2. previously more simple methods for very similar purpose have been used by the main author, however, any comparison bethween the outputs is missing. This should be included in the paper as it would also give the readers suggestion on possible differences if there are any. 

3. For the estimates of activity scores any drugs with CYP2C19 dependent PK have been included, which is rather surprising as no consideration has been taken into account for the differences among the compounds with respect to their relative dependency of this metabolic pathway. This clearly underestimates the scores for compounds which depend on the pathway the most, while overestimets the relatively less dependent. For sure this approach does not allow precise estimates for any compounds. This should be implemented into the calculations. 

Author Response

Please see the attachment "ResponseReviewer 2"

Round 2

Reviewer 2 Report

Thank you for the revision, I have no further coments.